# An integrative transcriptomic atlas of organogenesis in human embryos

Dave T Gerrard[1†], Andrew A Berry[1†], Rachel E Jennings[1,2], Karen Piper Hanley[1], Nicoletta Bobola[3], Neil A Hanley[1,2*]

[1]Division of Diabetes, Endocrinology & Gastroenterology, School of Medical Sciences, Faculty of Biology, Medicine and Health, Manchester Academic Health Science Centre, University of Manchester, Manchester, United Kingdom; [2]Endocrinology Department, Central Manchester University Hospitals NHS Foundation Trust, Manchester, United Kingdom; [3]Division of Dentistry, School of Medical Sciences, Faculty of Biology, Medicine and Health, Manchester Academic Health Science Centre, University of Manchester, Manchester, United Kingdom

**Abstract** Human organogenesis is when severe developmental abnormalities commonly originate. However, understanding this critical embryonic phase has relied upon inference from patient phenotypes and assumptions from in vitro stem cell models and non-human vertebrates. We report an integrated transcriptomic atlas of human organogenesis. By lineage-guided principal components analysis, we uncover novel relatedness of particular developmental genes across different organs and tissues and identified unique transcriptional codes which correctly predicted the cause of many congenital disorders. By inference, our model pinpoints co-enriched genes as new causes of developmental disorders such as cleft palate and congenital heart disease. The data revealed more than 6000 novel transcripts, over 90% of which fulfil criteria as long non-coding RNAs correlated with the protein-coding genome over megabase distances. Taken together, we have uncovered cryptic transcriptional programs used by the human embryo and established a new resource for the molecular understanding of human organogenesis and its associated disorders.

*For correspondence: neil.hanley@manchester.ac.uk

[†]These authors contributed equally to this work

**Competing interests:** The authors declare that no competing interests exist.

## Introduction

Embryogenesis encompasses the progression from fertilized zygote to blastocyst and through gastrulation to establish the three germ layers of ectoderm, mesoderm and endoderm, from which all organs and tissues subsequently arise during organogenesis. Remarkably little is known about this latter phase of assembling organs and tissues in human due to the restricted availability of human embryonic tissue and its tiny size. Previous transcriptomics post-implantation have sampled either the whole embryo by expression microarray (*Fang et al., 2010*), thus lacking organ-specific resolution and the vast majority of long non-coding (lnc) transcription; or included lnc expression by massively parallel short-read RNA sequencing (RNA-seq) but focussed on single sites such as limb bud (*Cotney et al., 2013*) or pancreas (*Cebola et al., 2015*). RNA-seq from NIH Roadmap and other studies during or after the end of the first trimester of pregnancy falls after the embryonic period (which ends at 56–58 days post-conception (Carnegie Stage 23)) and commonly reflects near terminal differentiation within heterogeneous fetal organs and tissues (*Jaffe et al., 2015*; *Roadmap Epigenomics Consortium, 2015*; *Roost et al., 2015*). As a consequence of these combined deficiencies, we set about compiling global transcriptomic data during the critical phase of human organogenesis, sampling each germ layer and including sites of mixed origin that are subject to major developmental disorders such as cleft palate and limb abnormalities (*Figure 1a*).

**eLife digest** Individual organs and tissues form in human embryos during the first two months of pregnancy. Any errors during this crucial stage of human development can result in miscarriage or serious birth defects. Yet remarkably little is known about how this process works. What is known has been inferred from studies of how other animals develop, human stem cells grown in a laboratory, and babies born with genetic conditions that cause developmental problems.

Genes control the way that organs and tissues form, and are switched on or off in complex patterns during development to ensure that particular cells develop into one type of organ and not another. When genes are switched on, their DNA is copied into molecules called RNA. Many RNA molecules are used as templates to make proteins, which then perform critical roles in cell processes. One way to find out which genes are activated during development is to identify which RNAs are made by cells in the embryo.

Here, Gerrard, Berry et al. used a technique called RNA-sequencing to identify the RNAs that human embryos make while their organs and tissues form. The RNA came from many different tissues including the heart, limbs and the roof of the mouth. Gerrard, Berry et al. developed a new computational model that used the identity of the RNAs to decode the precise patterns of gene activity in the tissues. The model correctly identified many genes that were already known to cause developmental problems when faulty, and identified numerous others that are now predicted to cause developmental defects in humans.

Gerrard, Berry et al. also discovered over 6,000 RNAs in the human embryos that are unlikely to code for proteins. These "non-coding" RNAs may have other roles in cells, such as switching off genes, and many of them appear to be specific to human embryos. Together, these findings have uncovered new patterns of gene activity that drive development in human embryos and provide a resource for studying how organs and tissues form. Future challenges are to understand what controls these patterns of gene activity, and how the patterns change over time.

Organs and tissues from fifteen human embryonic sites were sequenced in two sets of biological replicates (except pancreas and tongue) to generate 28 strand-specific RNA-seq datasets with 44–90 million uniquely mapped reads per replicate (*Figure 1a*; *Supplementary file 1A*, which contains information on embryonic stages). Global transcription rates across all organs and tissues were comparable over a high dynamic range; approximately 70% of protein-coding genes contained 100–10,000 mapped reads (*Figure 1b*; *Supplementary file 2*). We assessed whether our human embryo datasets identified earlier developmental processes than currently available fetal data (*Roadmap Epigenomics Consortium, 2015*). There were three-fold the number of differentially expressed genes in the fetal datasets but equivalent enrichment of gene ontology (GO) terms in the embryo, including many early developmental processes such as morphogenesis of an epithelial bud, anterior/posterior pattern specification and embryonic morphogenesis. These were in contrast to homeostatic processes enriched in the fetal dataset (*Figure 1c*; *Supplementary file 1B–C*).

Sampling gene expression across multiple sites allowed us to set about deciphering the precise transcriptomic codes responsible for the development of the different human embryonic organs and tissues. While *ZNF* and *ZSCAN* family members were broadly expressed discrete site-specific expression was more apparent for individual members of other transcription factor families (*Figure 1—figure supplement 1*) exemplified by the *HOX* gene clusters (*Figure 1d*). User-defined sets of up to five developmental transcription factors characteristic for a particular organ or tissue displayed very high levels of tissue specificity (*Figure 1—figure supplement 2*). However, while principal components (PC) analysis (PCA) or clustering grouped biological replicates, relationships between different organs and tissues other than the distinctiveness of brain and liver were not resolved (*Figure 1—figure supplements 3–4*). Non-negative matrix factorisation (NMF) also allows unbiased clustering of gene expression (*Gaujoux and Seoighe, 2010*). By setting the parameters such that representative genes were only extracted once, we identified eleven non-overlapping 'metagenes' from the complete expression dataset with clear tissue-specific signals for thyroid, liver, RPE, brain, heart and adrenal gland (*Figure 1—figure supplement 5*; *Supplementary file 1D*). We hypothesized that

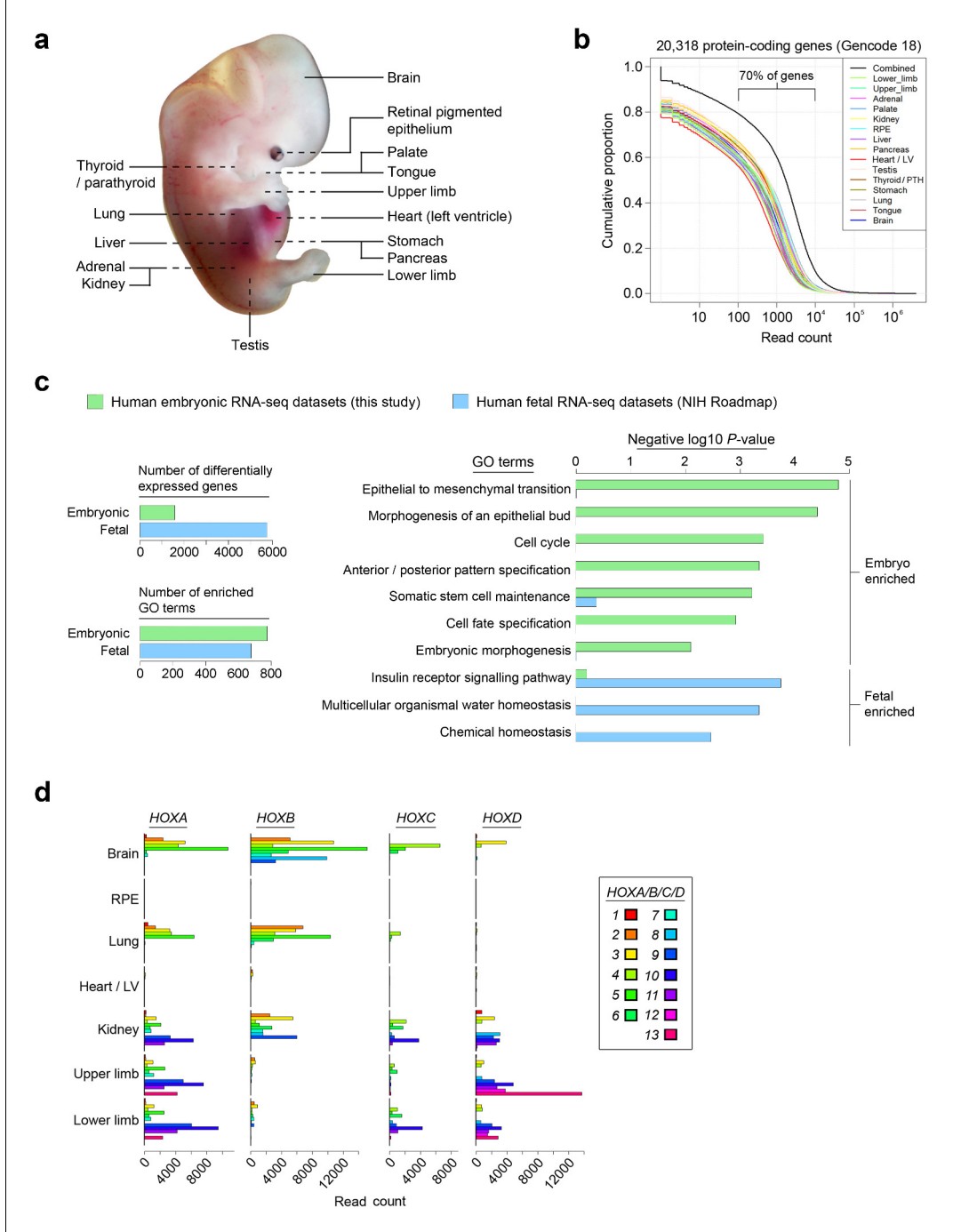

**Figure 1.** Profiling the transcriptomes underlying organogenesis in human embryos. (**a**) Human embryo showing the 15 tissues and organs subjected to RNA-seq. (**b**) High dynamic range of human embryonic RNA-seq. The combined dataset (black) included expression of >90% of annotated protein-coding genes (GENCODE18 [***Harrow et al., 2012***]). (**c**) Human embryogenesis possesses a distinctive transcriptome. Human embryonic read counts were compared with equivalent fetal datasets from NIH Roadmap (***Roadmap Epigenomics Consortium, 2015***) using edgeR (***Robinson et al., 2010***) and a false discovery rate (FDR) of 0.05 (see Materials and methods, ***Supplementary file 1B***). Negative log10 p-values are shown for selected biological process Gene Ontology (GO) terms with significant enrichment in either the embryonic or fetal gene sets following Fisher's exact test applied using the elimination algorithm (***Alexa and Rahnenfuhrer, 2010***) (***Supplementary file 1C*** contains the full list of enriched terms). (**d**) Selected sites illustrate the highly specific expression of *HOX* genes within the human embryo.

*Figure 1 continued on next page*

*Figure 1 continued*

The following figure supplements are available for figure 1:

**Figure supplement 1.** Transcription factor atlas of human organogenesis.

**Figure supplement 2.** Heatmap of user-defined transcription factors indicates organ and tissue specificity during human organogenesis.

**Figure supplement 3.** Principal components analysis of the human embryonic transcriptomes.

**Figure supplement 4.** Heatmap of RNA-seq samples.

**Figure supplement 5.** NMF Metagene analysis.

---

these new signals might allow benchmarking to assess the fidelity of in vitro differentiated stem cells, similar to a previous report (*Roost et al., 2015*). As an exemplar, we chose hepatocyte differentiation for which RNA-seq data are available including positive (primary adult hepatocytes) and negative (human embryonic fibroblasts) control data (*Du et al., 2014*). Clear enrichment for the stem cell-derived hepatocytes and the primary hepatocytes (but not the fibroblasts) was apparent in metagene 2, the cluster of 39 genes indicative of human embryonic liver. From this starting point, we wanted to move beyond the unique organ-specific signatures to study how patterns of gene expression co-varied across tissues. While relaxing NMF parameters would allow non-exclusive gene selection across metagenes, we also wanted to capture differences in gene expression between organs (e.g. aspects of what is not expressed as a contributing factor to an organ's identity). Moreover, different embryonic organs are related according to developmental lineage. We reasoned that being able to apply a lineage structure would create natural assemblies of co-regulated genes (*Figure 2a*). Accordingly, we adapted a method from spatial ecology and phylogenetics (*Jombart et al., 2010a, 2010b*) to constrain PCA by imposing a hierarchical developmental lineage and termed this approach LgPCA. We also included RNA-seq from undifferentiated human embryonic stem cells (*Roadmap Epigenomics Consortium, 2015*) to represent pre-gastrulation human biology. Of the total thirty-one principal components (PCs) arising from LgPCA the first fifteen now identified patterns of gene expression across groups of related tissues in addition to unique organ-specific signatures (*Figure 2b*) while PCs 16–31 sampled heterogeneity within individual organs and tissues (*Figure 2—figure supplement 1*). In keeping with this transition PCs 1–4 ordered samples reminiscent of very early developmental events: pluripotency (extreme positive loadings in PC1; 'PC1 high'), early brain formation (extreme negative loadings in PC2; 'PC2 low'), foregut endoderm (PC4 low) and intermediate mesoderm (PC4 high). PCs 5–15 resolved the individual organs and tissues; for instance low PC5 loadings discriminated liver from the other foregut endoderm derivatives. Heatmaps illustrated the composite or tissue-specific signals emanating from the genes with the most extreme PC loadings which also underlay appropriate developmental gene ontology (GO) terms (*Figure 2c–d* and *Supplementary file 1E–F*).

Identifying the master regulators that differentially orchestrate organogenesis across the body has not previously been possible directly in human embryos. We undertook this in two different ways, both based on studying the 1000 genes with the most extreme loadings in PCs 1–15 that identified gene co-expression patterns across tissues and within individual organs (*Figure 2b*; *Supplementary file 1E*). We interrogated these gene sets for regulatory networks based on the enrichment of transcription factor motifs (*Janky et al., 2014*). Numerous well known master regulators were recovered alongside previously unappreciated factors for either broad tissue groups (e.g. foregut endoderm derivatives) or individual organs (*Figure 3a*). As proof-of-principle, this also included proven regulators of human pluripotency, NANOG, OCT4 and MYC, at an extreme of PC1. Remarkably, in several instances approaching half of the 1000 genes with the most extreme PC loadings imputed co-regulation by a single transcription factor, such as HNF4A in the liver or SRF in the heart. Alongside NR5A1, the data predicted RUNX and BAD as novel regulators of human adrenal and gonadal development (*Figure 3a*). As a second approach to study gene regulation, we extracted the transcription factors (typically ≤5%) from amongst the 1000 most extreme genes in

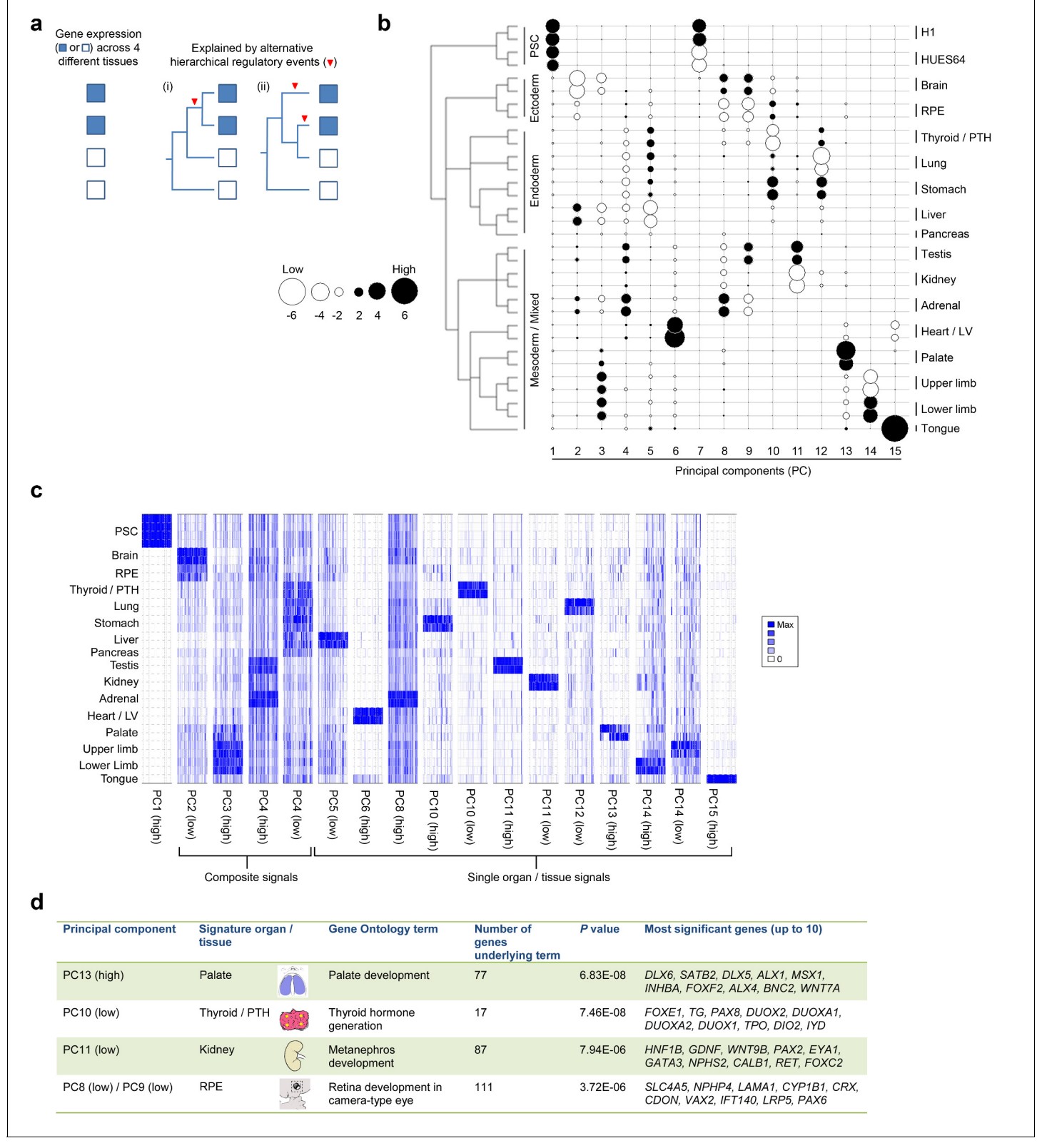

**Figure 2.** Lineage-guided PCA discovers unique transcriptional signatures regulating human organogenesis. (**a**) Interpreting gene expression profiles is dependent upon the underlying developmental lineage. Similar expression profiles in closely related tissues imply fewer regulatory events. (**b**) Lineage-guided principal components analysis (LgPCA) constrains PCA by imposing a developmental lineage on the different organs and tissues. The first 15 PCs are shown including biological replicates for the human embryonic organs and tissues integrated with human embryonic stem cell data

*Figure 2 continued on next page*

*Figure 2 continued*

(**Roadmap Epigenomics Consortium, 2015**). PC scores for the 15 different dimensions are shown in black (positive/high) or white (negative/low) with scale (extremeness) indicated by circle size (sign/direction is arbitrary). Unique transcriptional signatures were resolved for broad organ groupings (e.g. foregut endoderm derivatives, low scores in PC4), single organs or tissues (e.g. palate, high scores in PC13) or across tissues unrelated by germ layer but connected by multisystem congenital disorders (e.g. heart and limb, low scores in PC13). (**c**) Heatmaps of quantile normalised expression values of the most extreme 50 genes for selected PCs from the LgPCA. (**d**) Gene Ontology (GO) terms and their underlying genes illustrate the specific signatures from the LgPCA (further examples in **Supplementary file 1F**).

The following figure supplement is available for figure 2:

**Figure supplement 1.** Lineage-guided principal components analysis (LgPCA) for all 31 PCs.

PCs 1–15 (**Supplementary file 1G**). We searched the Mouse Genome Informatics database (MGI) and in 309/594 instances there was a relevant mouse mutation phenotype supporting the notion that the transcription factors identified by LgPCA are key regulators of human organogenesis. At the

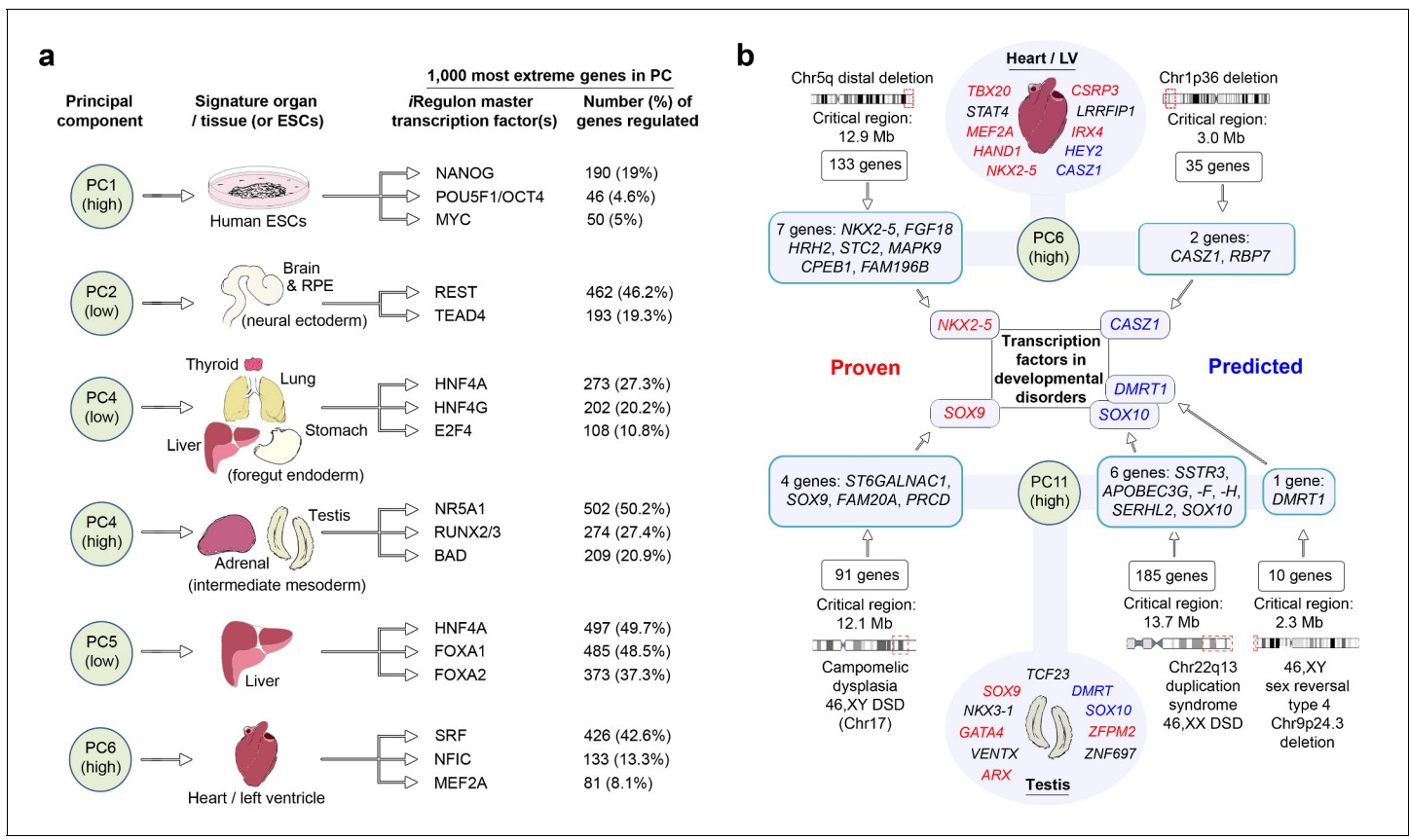

**Figure 3.** LgPCA points to master regulators of human organogenesis and the causes of human congenital disorders. (**a**) Predicted regulation by iRegulon (**Janky et al., 2014**) of the most extreme 1000 genes for different PCs identifies known and unexpected transcription factors regulating human organogenesis. In several examples, individual transcription factors (e.g. REST, NR5A1, HNF4A, FOXA1 and SRF) were predicted to regulate nearly half of the most extreme 1000 genes. (**b**) Transcription factors at the extremes of individual PCs in the LgPCA are responsible for a diverse range of congenital disorders (red names in the ovals for heart and testis; full details in **Supplementary file 1G**). To validate the utility of these data, we conservatively selected some of the earliest critical regions for these disorders (two 'Proven' examples on the left; all 53 listed in **Supplementary file 1H**). LgPCA frequently isolated the correct transcription factor from an average of 111 genes across >10 Mb, shown for NKX2-5 in congenital heart disease and SOX9 in campomelic dysplasia. Beyond this validation LgPCA similarly predicts causative transcription factors (blue) for many unresolved congenital disorders such as developmental heart abnormalities in Chr1p36 deletion syndrome and sex reversal / disorders of sex differentiation (DSD) (all 13 examples in **Supplementary file 1H**).

lowest extreme of PC5 (liver) the twenty-two transcription factors contained all three of those required for reprogramming fibroblasts directly to hepatocytes (*Huang et al., 2014*). This suggests novel fate programming roles for transcription factors at the extremes of other PCs (including new potential regulators of pluripotency amongst the sixteen factors containing zinc fingers in PC1). In keeping with these regulatory roles, the extreme PC loadings in the LgPCA data also prioritized those transcription factors responsible for major congenital disorders (*Supplementary file 1G*). Because LgPCA is not limited to individual organs this included a novel ability to predict multisystem abnormalities such as the combined heart and limb defects of Holt-Oram syndrome (OMIM 142900, *TBX5*, PC13 low) or the palate and limb abnormalities associated with mutations in *TP63* (OMIM 603543, PC3 high).

Mutations in genes encoding transcription factors are over-represented causes of congenital disorders, most likely due to their critical function during organogenesis and inadequacy when haploinsufficient. The enrichment of transcription factors with specific disease-associations at the extremes of the LgPCA implicates the co-enriched genes as leading candidates for unanswered clinical syndromes. To test this model we identified some of the earliest chromosomal mapping or patient deletion data for the known disease-associated transcription factors from *Supplementary file 1G*. 53 disorders were suitable for assessment with an average critical region of 13.7 Mb each containing an average of 111 protein-coding genes (*Supplementary file 1H*). Strikingly, in 37 instances (73%) LgPCA uniquely selected the correct transcription factor and in 48 instances (91%) narrowed the field down to three or fewer transcription factors. When applied to 13 syndromes (mostly deletion disorders) where the causative gene remains unresolved clear predictions of causality emerge, for instance in cleft palate (*DLX5*, *DLX6*, *LHX8* and *FOXF2*) or cerebellar disorders (*ZIC1* and *ZIC4*) (*Supplementary file 1H*). Frequently, there is an appropriate mutant mouse phenotype such as *CASZ1* in cardiac malformations, part of Chr1p36 deletion syndrome, or *SOX10* in the 46,XX disorder of sex differentiation (DSD) linked to duplication on Chr22 (*Figure 3b* and *Supplementary file 1H*).

Non-coding transcription has emerged as a critical regulator of cell and developmental biology (*Goff and Rinn, 2015*). A dedicated programme operating during human organogenesis seemed likely as 81 out of the 1571 genes enriched in embryogenesis compared to the fetal datasets were annotated long intergenic non-coding (LINC) transcripts (*Supplementary file 1B*). To look beyond this we assembled strand-specific transcripts not recognized by current genome annotation [GENCODE 18 (*Harrow et al., 2012*)] and systematically named them individually according to recommended criteria (*Mattick and Rinn, 2015*). 6251 unique loci accounted for in excess of 9 Mb of novel polyadenylated transcription from the human genome (*Figure 4a* and *Supplementary file 1I*). The vast majority of transcripts fulfilled criteria as lnc RNAs by assessment of coding potential (CPAT score <0.2) (*Figure 4b*), length over 200 base pairs (bp) and an absence of reads spanning splice junctions to currently annotated genes (*Mattick and Rinn, 2015*). These lncRNAs were classified as either bidirectional, antisense or overlapping, or by exclusion intergenic, according to orientation and position in relation to the annotated genome (*Mattick and Rinn, 2015*). Transcripts were most commonly 500–1,500 bp but could extend to over 600 Kb (*Figure 4c*) and showed high tissue-specificity with the median Tau value (*Yanai et al., 2005*) of 0.86, much higher than for protein-coding genes (0.63) but consistent with previously annotated non-coding genes (0.89). We investigated the association between this novel human embryonic transcriptome and the annotated genome. Reduced physical distance to expressed annotated genes markedly increased the likelihood of novel transcript co-expression (*Figure 4e*), although the best correlations were by no means always with the closest gene (*Figure 4f–g*). The median distance to the closest annotated gene was 7.7 Kb (*Figure 4—figure supplement 1*) while on average the best correlation was at 188 Kb (random prediction was 476 Kb). Over half (3634) of the lnc transcripts were classified positionally as LINC RNAs. While LINC RNAs can harbour important regulatory function, how to forecast their relationship(s) with the protein-coding genome and prioritize the investigation of thousands of new transcripts is immensely challenging (*Goff and Rinn, 2015*). As a first step, the multi-tissue nature of our dataset allowed intricate correlative patterns to be deciphered implying putative relationships; for instance over a 2 Mb window and across numerous genes on chromosome 7 between *HE-LINC-C7T121* and *TBX20*, which encodes a developmental cardiac transcription factor mutated in a wide range of congenital heart disease (*Figure 4h*).

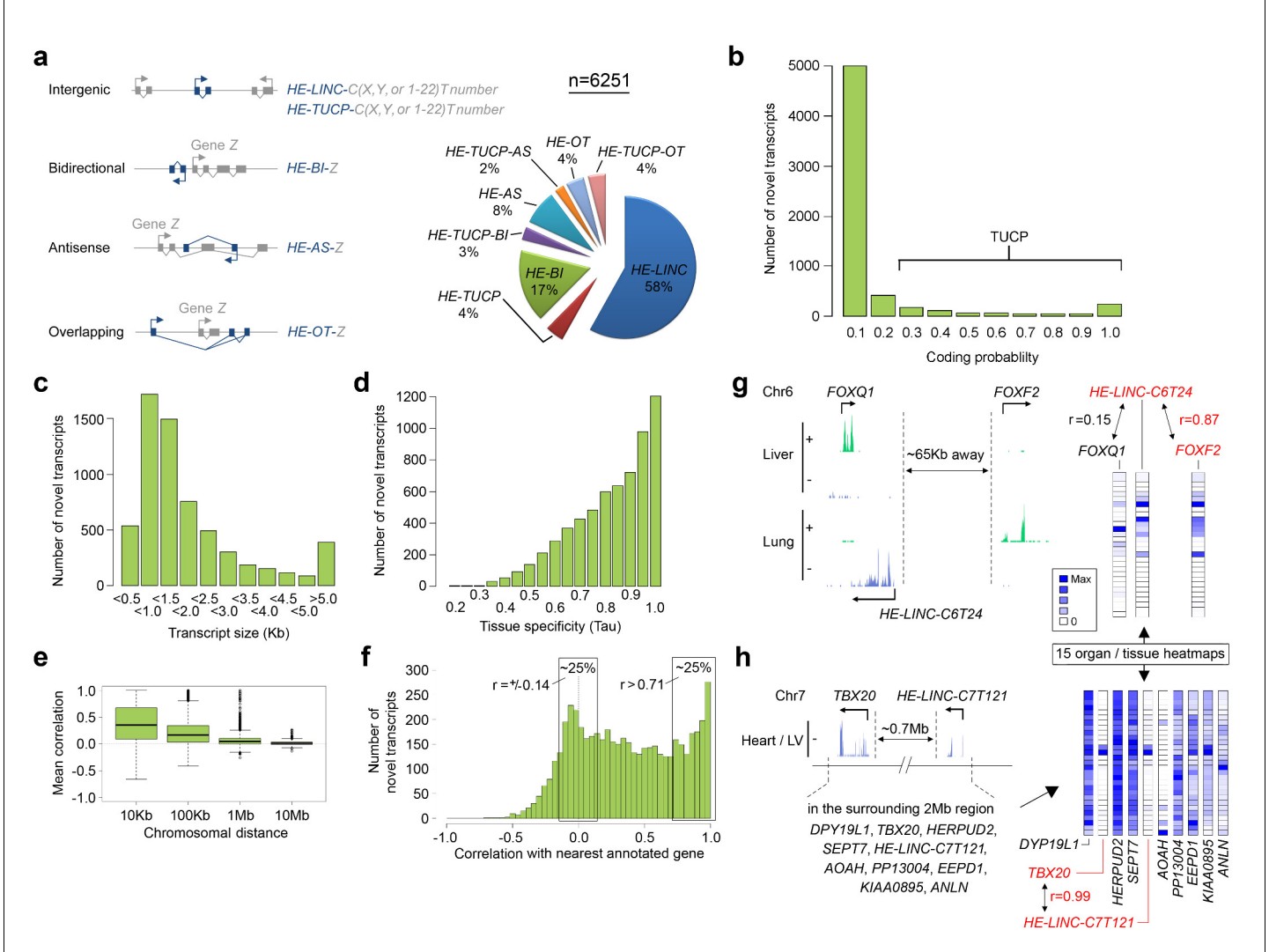

**Figure 4.** 6251 novel transcripts identified during human organogenesis show low coding probability and high tissue-specificity. (**a**) Novel transcript models were merged across tissues (n = 9180; *Supplementary file 4*), assessed for coding potential using CPAT and classified (*Mattick and Rinn, 2015*) as overlapping (OT), antisense (AS), bidirectional (BI), intergenic noncoding (LINC) and/or transcripts of uncertain coding potential (TUCP, if CPAT >0.2). LINC or TUCP transcripts were numbered sequentially (T number) along each chromosome (C, either X, Y or 1–22) whereas BI, AS and OT transcripts were named by association with the annotated gene ('Z'). A small proportion of transcripts fulfilled dual criteria as BI/AS/OT and TUCP. 6251 unique, non-overlapping, filtered transcript models were identified (the longest from each locus, >200 bp; *Supplementary file 1I*). (**b**) Histogram of coding probability determined using CPAT (*Wang et al., 2013*). 9% of transcripts were classed as TUCP. The small proportion with clear open reading frames (CPAT score = 1.0) were predominantly OT transcripts. (**c**) Distribution by size of transcript. 114 transcripts were >10 Kb. (**d**) Tissue specificity was calculated using Tau (*Yanai et al., 2005*) based on the mean normalized read counts for each tissue or organ site. 80% of transcripts showed Tau values >0.7 indicating high tissue specificity. Details on exon and read counts, and proximity to surrounding genes are shown in *Figure 4—figure supplement 1*. (**e**) Box and whisker plots show the correlation between expression of the novel transcripts and surrounding annotated genes within set chromosomal distances of the novel transcriptional start site. Mean correlation was near zero beyond 1 Mb. (**f**) Histogram showing the correlation (r) between expression of each novel transcript and its closest annotated gene. One quarter of novel transcripts show a correlation (r > 0.71) with the nearest gene; another quarter shows minimal correlation (r = ±0.14). There was no strong anticorrelation. g-h, Expression of the novel transcript is not always correlated with the immediately adjacent gene, illustrated by heatmaps across the 15 organs and tissues. (**g**) Expression of the novel transcript, *HE-LINC-C6T24*, located just over 2 Kb from *FOXQ1*, correlates strongly with *FOXF2*, approximately 65 Kb distant. (**h**) Heatmap demonstrates the poor correlation of expression between *HE-LINC-C7T121* and most of the nine genes within 1 Mb on Chr7 but near perfect correlation with *TBX20* located ~0.7 Mb away beyond two intervening genes.

The following figure supplement is available for figure 4:

**Figure supplement 1.** Exon and read counts and distance to the nearest annotated gene for the novel human embryonic transcripts.

Taken together, this study reports the first comprehensive transcriptomic atlas during human organogenesis to complement parallel initiatives from later development and adulthood (*Jaffe et al., 2015*; *Roadmap Epigenomics Consortium, 2015*; *Roost et al., 2015*). Subjecting transcription from many sites to a method of analysis that incorporated developmental lineage deciphered novel genetic signatures, predicted causality in many human developmental disorders and associated novel non-coding transcription with expression from the surrounding protein-coding genome. At present, the data arise from a relatively narrow window of embryonic development but set the stage for future longitudinal studies for individual organs over time. The tiny amounts and scarcity of human embryonic tissue also necessitated aspects of pooling across different Carnegie stages for some sites but it is striking that this had no impact on ascertaining organ and tissue-specific transcriptomic signatures by LgPCA. The integrated data are expected to be particularly valuable to stem cell researchers examining the fidelity of PSC differentiation in vitro or searching for transcription factors for direct reprogramming of chosen cell lineages. Finally, the discovery of a major new programme of non-coding transcription adds a fresh layer of detail on the spatiotemporal regulation of the human genome.

## Materials and methods

### Human material

Human embryonic material was collected under ethical approval, informed consent and according to the Codes of Practice of the Human Tissue Authority and staged by the Carnegie classification as described previously (*Jennings et al., 2013*). This clinical material was collected on site overseen by our research team with immediate transfer to the laboratory. Individually identified tissues and organs (details in *Supplementary file 1A*) were immediately dissected. The adrenal gland, whole brain, heart, kidney, liver, entire limb buds, lung, stomach, testis, thyroid and anterior two-thirds of the tongue were readily identifiable as discrete organs and tissues. All visible adherent mesenchyme was removed from organs and tissues under a dissecting microscope. For the adrenal gland, this includes the capsule which allowed separation from the kidney. The ureter, emerging from the renal pelvis, was removed separately from the kidney. For the heart, a window of tissue was removed from the lateral wall of the left ventricle. A segment of the liver was dissected from each embryo that avoided the developing gall bladder. The trachea was removed from the lung at its entry point into the lung parenchyma. The stomach was identified between the gastro-oesophageal and pyloric junctions. The testis was dissected free from the attached mesonephros. While the thyroid was readily visualized as a discrete organ in the neck, it unavoidably contained the developing parathyroids and thus this tissue type was referred to throughout as 'thyroid/PTH'. The palatal shelves were dissected on either side of the midline. The eye was dissected and the RPE peeled off mechanically from its posterior surface with validation possible under the dissecting microscope because the RPE is very darkly pigmented compared to the other ocular structures. All samples were collected into Trizol (Thermofischer) or Tri reagent (Sigma-Aldrich) for total RNA isolation as individual tissue or organ types followed by treatment with DNaseI (Sigma-Aldrich). Once the quality of each RNA sample had been confirmed, samples were pooled in order to obtain sufficient RNA for each biological replicate (*Supplementary file 1A*). The pancreas dataset derived from the same tissue collection was re-used from a previous study (*Cebola et al., 2015*).

### RNA-seq and transcriptome

Quality and integrity of total RNA samples were assessed using a 2100 Bioanalyzer or a 2200 TapeStation (Agilent Technologies) according to the manufacturer's instructions. RNA sequencing (RNA-seq) libraries were generated using the TruSeq Stranded mRNA assay (Illumina, Inc.) according to the manufacturer's protocol. Briefly, total RNA (0.1–4 μg) was used as input material from which polyadenylated mRNA was purified using poly-T, oligo-attached, magnetic beads. The mRNA was then fragmented using divalent cations under elevated temperature and then reverse transcribed into first strand cDNA using random primers. Second strand cDNA was then synthesized using DNA Polymerase I and RNase H. Following a single 'A' base addition, adapters were ligated to the cDNA fragments, and the products purified and enriched by PCR to create the final cDNA library. Adapter indices were used to multiplex libraries, which were pooled prior to cluster generation using a cBot

instrument. The loaded flow-cell was then sequenced (paired-end; 101 + 101 cycles, plus indices) on an Illumina HiSeq2000 or HiSeq2500 instrument. Demultiplexing of the output data (allowing one mismatch) and BCL-to-Fastq conversion was performed with CASAVA 1.8.3. The RNA-seq was conducted in three batches at different times as a necessity of how human embryonic tissue was collected over time (*Supplementary file 1A*). Where organs were sequenced across batches (palate, RPE, kidney, testis, adrenal gland, heart / left ventricle and liver) biological replicates clustered together (*Figure 1—figure supplement 4*).

RNA-seq reads from the Illumina platform were mapped to the human genome (hg19) strand-specifically using TopHat 2.0.9 (*Trapnell et al., 2012*) and the GENCODE 18 gene annotation set (*Harrow et al., 2012*). We also remapped the published pancreas RNA-seq dataset (*Cebola et al., 2015*) obtained from material isolated previously in our laboratory. Additionally, a dataset of hepatocyte differentiation RNA-seq (*Du et al., 2014* GEO: GSE54066) was downloaded, re-mapped and quantified as per our own data. Commonly applied RNA-seq normalisation methods such as TMM assume a small proportion of differentially expressed genes in any one dataset (*Dillies et al., 2013*). Because the highly distinct tissues surveyed here differed strongly on the scale of thousands of genes (for instance liver versus brain) we used quantile normalisation which gave a lower median coefficient of variation than either no or TMM normalization. Read counts from the different datasets were quantile normalized using the R package preprocessCore (*Bolstad, 2007*). Tissue-specificity was scored per gene using Tau (*Yanai et al., 2005*) on normalized read counts across all samples. Initial genome-wide relationships were assessed using PCA (*Figure 1—figure supplement 3*) and hierarchical clustering (heatmap, *Figure 1—figure supplement 4*).

To compare our samples with RNA-seq from the NIH Roadmap project (*Roadmap Epigenomics Consortium, 2015*) uniquely mapped strand-specific RNA-seq reads were counted into a set of non-redundant exon annotations (custom made from GENCODE 18 annotations) using bedtools intersect (*Quinlan and Hall, 2010*). Exon level counts were then summed into a single total per gene per sample. Counts were quantile normalized across samples. NIH roadmap samples (*Roadmap Epigenomics Consortium, 2015*) used in this study are listed in *Supplementary file 1J*. For the analysis of human embryonic RNA-seq with comparable Roadmap fetal data (adrenal gland, heart, kidney, lung, limbs, stomach and testis) a single pairwise differential expression test was undertaken using the R package edgeR (*Robinson et al., 2010*) and an FDR $\leq$ 0.05.

## NMF

Non-negative matrix factorisation (NMF) searches complex expression data, comprising thousands of genes, for a small number of characteristic 'metagenes' (*Gaujoux and Seoighe, 2010*). NMF was performed using the nmf R package (version NMF_0.20.5) (*Gaujoux and Seoighe, 2010*) to extract tissue-specific metagenes. Non-normalised read counts were filtered to remove all Y-linked genes, the X-inactivation gene *XIST* and genes with fewer than 100 reads across all samples. Initially 50 runs each of ranks 11–18 and using the default 'Brunet' algorithm (*Brunet et al., 2004*) were performed to find an optimal factorisation 'rank' (r). The maximal cophenetic distance was used to select the value of r. Subsequently, 200 runs using the optimal rank were performed to assess consistency of sample groupings between runs. Non-overlapping (i.e. tissue-specific) gene sets were extracted from each metagene by filtering on basis contribution $\geq$0.8.

## LgPCA

The LgPCA approach was adapted from established phylogenetic PCA methodology (*Jombart et al., 2010b*) and performed using quantile-normalized, gene-level read counts, a high memory (512 Gb) compute node and the ppca function from the adephylo R package (*Jombart et al., 2010a*). A broad user-defined guide tree (*Figure 2b*) based on well-established knowledge of mammalian gastrulation and downstream lineage relationships was imposed on the different organ and tissue types following which the adephylo R package weighted the principal components by the lineage auto-correlation between samples; increased if related samples were similar and lessened if related samples were more different. As in the description from Jombart and colleagues the resulting components represented 'global' structures (where similarity is high between related samples) and 'local' structures (where related samples are dissimilar) (*Jombart et al., 2010b*). We used the LgPCA to extract all the global patterns from the data (PCs

1–15). These patterns were not apparent if lineage relationships were not included nor were they altered if any one tissue, such as palate, was altered within the broad lineage structure (data not shown). The global patterns in PCs 1–15 infer co-regulatory patterns of gene expression across human organogenesis. The 'local' patterns thereafter captured heterogeneity between tissue replicates (*Figure 2—figure supplement 1*) (while PC7 separated the two PSC populations these RNA-seq datasets represent separate cell lines from NIH Roadmap). We used the Abouheif distance as implemented in adephylo (*Jombart et al., 2010a*), which takes into account the topology of the specified tree but does not use branch lengths.

## Gene set enrichment

For the comparison of the embryonic versus fetal datasets Gene Ontology term enrichment was performed on upregulated genes (FDR $\leq$ 0.05) using Fisher's exact test with the elimination algorithm of the R package topGO (*Alexa and Rahnenfuhrer, 2010*). For the LgPCA, annotated ontology nodes (>10 genes) were tested for each loadings vector for each PC against background using the Wilcoxon test. Tests were performed sequentially moving up the separate GO ontologies (Biological Process (BP), Molecular Function (MF) and Cellular Component (CC)), excluding significant scoring genes from later tests (the topGO 'elim' method).

## iRegulon analysis of regulation in the extremes of the LgPCA

iRegulon is a computational method which tests for enrichment amongst precomputed motif datasets to decipher transcriptional regulatory networks in a set of co-expressed genes. The 1000 genes with the most extreme loadings at either end of each PC ('high' and 'low') from the LgPCA were loaded into Cytoscape (version 3.2.1) (*Shannon et al., 2003*) and used as queries to the iRegulon plug-in (version 1.3, build 1024) (*Janky et al., 2014*). 20 Kb was examined centred on the transcriptional start site (TSS) under default settings.

## Novel transcripts

Sample-specific transcriptomes were assembled with Cufflinks (version 2.2.0) (*Trapnell et al., 2010*). Transcriptomes were combined ('cuffmerge'; -min-isoform-fraction = 0.1) and compared with the original GENCODE 18 reference ('cuffcompare'). We filtered out known transcripts using the 'Transfrag class codes' (http://cole-trapnell-lab.github.io/cufflinks/cuffcompare/#transfrag-class-codes) to retain only wholly intronic ('i', of which there were none), unknown ('u'), antisense (x) and overlapping ('o') transcripts. We discarded all other classes including pre-mRNA (class 'e'), novel-isoforms spliced to known exons (class 'j'), and 3' run-ons within 2 kb of the end of the transcript annotation (class 'p'). In addition, some remaining non-spliced transcripts may theoretically represent first or last exon (UTR) extensions; to delimit these, we calculated the distance on the same strand to the closest downstream transcription start site (to consider potential 5' UTR extension) and upstream transcription termination site (to consider potential 3' UTR extension). Names were assigned to these novel transcripts following suggested criteria (*Mattick and Rinn, 2015*) (*Figure 4a*) with the sole adaptation that bidirectional (BI) transcripts were defined as having their TSS within 1 Kb of the TSS of the associated annotated gene. No transcripts mapped to the same strand within the introns of any annotated gene excluding the possibility of unspliced transcripts from annotated genes being erroneously defined as novel transcripts. All transcript sequences (annotated and unannotated) were scored for protein-coding potential using CPAT (based on human training data included with CPAT) (*Wang et al., 2013*). A threshold of >0.2 was used to define 'Transcripts of Uncertain Coding Potential' (TUCP). Where there were multiple transcripts from a single locus, the longest transcript was retained in assembling the final dataset of 6251 novel transcripts. Transcript level read counts for the embryonic samples and NIH Roadmap samples (*Supplementary file 1J*) were generated for the merged transcriptome using bedtools multicov (vers 2.21.0) (*Quinlan and Hall, 2010*). The correlations between each of the 6251 transcripts and all annotated genes within 1 Mb were calculated using only the human embryonic data from this study.

## Data availability

Mapping coordinates against multiple genome versions using a range of common pipelines and summary count data are hosted at www.manchester.ac.uk/human-developmental-biology. To view

data in the UCSC genome browser, a trackhub is available: http://www.humandevelopmentalbiol-ogy.manchester.ac.uk/data/hub_manchester_hdb/hub.txt.

## Acknowledgements

We are very grateful to all women who consented to take part in our research programme and for the assistance of research nurses and clinical colleagues at Central Manchester University Hospitals NHS Foundation Trust. We thank Ian Donaldson, Peter Briggs and Andy Hayes of the Bioinformatics and Genomic Technologies Core Facilities at the University of Manchester for assistance with RNA-sequencing. REJ is a UK Medical Research Council (MRC) clinical research training fellow. NAH is a Wellcome Trust senior fellow in clinical science (WT088566MA). This project received support from the Wellcome Trust (WT097820) with additional support from MRC project grants MR/L009986/1 to NB and NAH, the British Council (14BX15NHBG) to NAH and MR/J003352/1 to KPH.

## Additional information

### Funding

| Funder | Grant reference number | Author |
|---|---|---|
| Wellcome Trust | WT088566MA | Dave T Gerrard<br>Andrew A Berry<br>Neil A Hanley |
| British Council | MR/J003352/1 | Karen Piper Hanley |
| Medical Research Council | MR/L009986/1 | Nicoletta Bobola<br>Neil A Hanley |
| Wellcome Trust | WT097820MF | Neil A Hanley |
| British Council | 14BX15NHBG | Neil A Hanley |

The funders had no role in study design, data collection and interpretation, or the decision to submit the work for publication.

### Author contributions

DTG, Wrote the manuscript, Conducted the bioinformatics analyses, Devised the study and planned experiments, Conception and design, Analysis and interpretation of data, Drafting or revising the article; AAB, REJ, Collected, dissected and prepared material, Devised the study and planned experiments, Conception and design, Acquisition of data, Drafting or revising the article; KPH, Involved in study design, Conception and design, Drafting or revising the article; NB, Wrote the manuscript, Conducted the analysis of developmental disorders, Devised the study and planned experiments, Conception and design, Analysis and interpretation of data, Drafting or revising the article; NAH, Wrote the manuscript, Guarantor, Devised the study and planned experiments, Conception and design

### Author ORCIDs

Dave T Gerrard, http://orcid.org/0000-0001-6890-7213
Neil A Hanley, http://orcid.org/0000-0003-3234-4038

### Ethics

Human subjects: Human embryonic material was collected under ethical approval, informed consent and according to the Codes of Practice of the Human Tissue Authority (protocols number 13/NW/0205).

## Additional files

### Supplementary files

• Supplementary file 1. Supplementary tables. (A) Samples used in this study. Details on the material derived from individual human embryos (each listed according to the Carnegie Stage (CS)) used in the biological replicates and the sequencing statistics for each sample. A conversion of Carnegie Stage to an approximate days post-conception is available in *Jennings et al. (2015)* (open access). Gene level read counts are available for download as a TSV file in *Supplementary file 2*. (B) Differential gene expression between paired embryonic and fetal RNA-seq data. The R package edgeR (*Robinson et al., 2010*) was used to test for differential gene expression between embryonic and fetal (*Roadmap Epigenomics Consortium, 2015*) datasets. Shared tissues were adrenal gland, heart, lung, stomach, kidney, upper limb, lower limb and testis. The table is sorted by FDR (column H) and can be filtered by log fold change (column E) to give embryo-enriched genes (negative values) or fetal-enriched genes (positive values). (C) Gene Ontology (GO) terms and the genes underlying them for embryonic vs.fetal (Roadmap) up-regulated genes. Genes up-regulated in embryonic tissues versus fetal tissues (edgeR, FDR $\leq$ 0.05, see *Supplementary file 1B*) were tested for GO term enrichment using Fisher's exact test and the elimination algorithm implemented in the R package topGO (*Alexa and Rahnenfuhrer, 2010*). Separate tests were run for embryo up-regulated and fetal up-regulated genes. The table is sorted by enrichment in embryonic genes. (D) Tissue-specific genes contributing to metagenes. All genes with relative basis contribution (across metagenes) greater than 0.8 are listed. (E) The most extreme 1000 genes (high and low) for all principal components (PC1-31) of the LgPCA. The dataset is derived from genes annotated in GENCODE18. Raw gene-level loadings for each principal component are available for download as a TSV file in *Supplementary file 3*. (F) Gene Ontology (GO) terms and the genes underlying them for organ and tissue-specific transcriptomic signatures from the extremes of the LgPCA. GO terms were identified as enriched in extreme scoring genes (annotated in GENCODE 18) in the principal components (PCs) of the LgPCA. Due to the very large number of terms returned at p<0.0001 by Wilcoxon test (the topGO 'elim' method, see Materials and methods) an illustrative selection are listed with raw gene-level loadings available for download in *Supplementary file 3*. (G) Transcription factors in the extremes of the LgPCA and their links to developmental morbidity. The most extreme 1000 annotated genes (GENCODE 18) of the LgPCA dataset were filtered for transcription factors based on KEGG and PHANTOM5 annotations and for read counts >500. To identify disease associations each gene was entered as a search term in OMIM (www.ncbi.nlm.nih.gov/omim) and in PubMed. Batch queries were undertaken at Mouse Genome Informatics (MGI, www.informatics.jax.org) with 'Mammalian phenotype' as the output. (H) LgPCA predictions of causal genes for critical regions in either solved or unsolved developmental disorders. Fifty-three developmental disorders (Column A, 'solved') with causally associated transcription factors identified in the appropriate transcriptomic signature of *Supplementary file 1G* were originally defined by critical regions (Column C with hyperlink). These critical regions were identified by searching OMIM and usually derived from mapping data on affected families or chromosomal deletions in affected patients. Larger critical regions were preferentially selected to test more meaningfully whether the LgPCA model could have pinpointed the causal gene based solely on transcriptomic signatures that involved an affected organ(s) or tissue(s) (Column B). The average critical region was 13.7 Mb (Column D) and contained an average of 111 protein-coding genes (Column E; identified from searching BIOMART on ENSEMBL). In 48/53 instances (91%), LgPCA narrowed the field down to three or fewer transcription factors and in 37 instances (73%) excluded all except the correct transcription factor. Therefore, the same approach was applied to 13 unsolved developmental disorders (mostly deletion syndromes) with predictions made in each case for any type of protein-coding gene (Column H) and transcription factor(s) (Column I). In many instances the transcription factor in Column I possesses an appropriate mutant mouse phenotype. (I) 6251 unannotated transcripts identified during human organogenesis. These are the 6251 novel and distinct transcripts underlying *Figure 4* of the main text, which also describes the transcript classification: Anti-sense (AS), Overlapping (OT), Bidirectional (BI), Long-intergenic non-coding (LINC) and / or Transcripts of uncertain coding potential (TUCP) (based on *Mattick and Rinn, 2015*). Intergenic transcripts are numbered sequentially within each chromosome. Exon lengths and starts (blocks) are recorded here in UCSC BED12 format. Correlations in expression profile were calculated for annotated genes with transcript transcriptional start sites situated within

1 Mb of the novel transcript TSS; the total number of genes in this window is listed. Columns AF-AT (organs and tissues) represent mean, quantile-normalised read counts across tissue replicates. Correlations (and distance) are shown for the closest, best correlated or best anti-correlated genes and were generated using only embryonic RNA-seq data. The pipeline to generate transcripts, distinguish them from previous annotations, name, characterise and filter is described in the Materials and methods. (J) NIH roadmap samples (*Kundaje et al., 2015*) used in this study.

• Supplementary file 2. Gene level non-normalised RNA-seq read counts by sample for 42,423 gene annotations in GENCODE18. Gene details are given plus the minimum, maximum, median and standard deviation of read counts. Additionally, tissue-specificity is scored using Tau (*Yanai et al., 2005*) where '0' is equally expressed across all organs and tissues and '1' indicates absolute specificity to one site.

• Supplementary file 3. LgPCA scores. Raw gene-level scores for each principal component of the LgPCA.

• Supplementary file 4. Unfiltered novel transcripts. Prior to filtering a total of 9180 transcripts were detected during human organogenesis that are not annotated in GENCODE 18. The 6251 transcripts summarised in *Figure 4* of the main text and listed in *Supplementary file 1I* (Excel file) are marked by column 'filter_score'. The data include quantile normalized read counts for all embryonic samples in this study and a mean for each organ or tissue type. In addition to the correlations in *Supplementary file 1I*, the distances to, and correlations with, both the closest upstream transcription termination site (TTS) and downstream transcription start sites (TSS) of adjacent, same-strand genes are given to aid filtering for potential 5' and 3' run-ons. For comparison, the NIH Roadmap sample (adult and fetal) with most reads for each novel transcript is listed separately.

• Supplementary file 5. Transcription factor atlas of human organogenesis (high resolution). High resolution version of *Figure 1—figure supplement 1* where each individual transcription factor is text searchable. (Please note that this opens at 25 x 200 inches / 64 x 508 cm).

### Major datasets

The following dataset was generated:

| Author(s) | Year | Dataset title | Dataset URL | Database, license, and accessibility information |
|---|---|---|---|---|
| Dave T Gerrard, Andrew A Berry, Rachel E Jennings, Karen Piper Hanley, Nicoletta Bobola, Neil A Hanley | 2016 | An integrative transcriptomic atlas of organogenesis in human embryos | http://www.ebi.ac.uk/arrayexpress/experiments/E-MTAB-3928/ | Publicly available at EBI ArrayExpress (accession no: E-MTAB-3928) |

The following previously published dataset was used:

| Author(s) | Year | Dataset title | Dataset URL | Database, license, and accessibility information |
|---|---|---|---|---|
| Santiago A Rodriguez-Segui, Andrew A Berry, Rachel E Jennings, Neil A Hanley, Jorge Ferrer | 2015 | TEAD and YAP regulate the enhancer network of human embryonic pancreatic progenitors | http://www.ebi.ac.uk/arrayexpress/experiments/E-MTAB-3061/ | Publicly available at EBI ArrayExpress (accession no: E-MTAB-3061) |

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
