## [Decision Letter]

Thank you for submitting your article "An integrative transcriptomic atlas of organogenesis in human embryos" for consideration by *eLife*. Your article has been reviewed by three peer reviewers, and the evaluation has been overseen by Janet Rossant as the Senior Editor and Reviewing Editor. One of the three reviewers has agreed to reveal their identity: Majlinda Lako.

The reviewers have discussed the reviews with one another and the Reviewing Editor has drafted this decision to help you prepare a revised submission. Given the nature of this contribution, we would like to consider it in the category of a Tools and Resources paper.

This paper provides transcriptional profiles from different tissues during early human embryogenesis and as such represents a significant and novel contribution. To date most of our knowledge on human embryogenesis is based on extrapolation from model animals. These new data will provide a resource to the community enabling more understanding of tissue differentiation and organogenesis. It will also help with identification of new disease genes and understanding developmental defects.

Essential revisions:

1) There are some important issues raised by two of the reviewers with regard to the statistical analysis of the data. These concerns are provided in detail below and must be addressed in the revised manuscript.

2) The linkage of the transcriptional data to developmental disorders is not well validated. There is considerable mouse developmental and phenotype data available in databases that could be mined to enhance the value of your findings and help determine whether the genes identified are truly key regulators of tissue development.

3) The claim that novel transcripts are likely lncRNAs was questioned and needs to be analyzed more carefully.

Statistical issues:

There are concerns of the methodology that would require further clarification:

For the NMF, while the implementation of the methods is likely to be appropriate, there may be a potential flaw in the interpretation of what the analysis may achieved. Regarding "non-overlapping" metagenes, by default, genes can be represented in multiple metagenes, based on the Brunet algorithm in the NMF R package (also the default algorithm). The Methods state that this algorithm has been used, so following on from that, the assertion that "non-overlapping metagenes" could be extracted from the complete dataset and the "potential of coordinated deployment of overlapping genes" is ignored may be incorrect. Furthermore, the critique that "NMF failed to discriminate transcriptional signatures for a number or organs or tissues, or discern the relationships between them… " may also apply to the outcome of lgPCA. Therefore, this is not a compelling argument for discarding the NMF.

The analysis could have been taken further to examine the enrichment of all the metagenes, similarly to the one for liver (Figure 1—figure supplement 3) for all the specific tissue metagenes (“clear tissue-specific signals for thyroid, liver, RPE, brain, heart and adrenal gland”), and a similar approach could be taken for the downstream functional analysis of lgPCA.

For lgPCA, while the interpretation of the results would be correct, there is confusion in the nomenclature of loading and scores, which are fundamental concepts in PCA. PC scores refer to the eigenvalues (magnitude of variance) which determine the separation between the samples (as can be gleaned from Figure 2). PC loadings are eigenvectors and describe the contribution of the variables (in this case genes) that causes the separation of the samples. Therefore, PC scores should rightly be PCA loadings (main text, third and fourth paragraphs). This confusion should be attended to. Additionally, it would be helpful to mention at least once in the main text that when referring to "PC1 low" for example, this is meant to be low/negative PC1 scores.

Both NMF and lgPCA are doing similar things, i.e. clustering samples based on gene expression to find which genes drive sample separation. The differences are that NMF can reveal the similarity between gene expression patterns for samples, which are clustered with the same metagenes, in this case kidney and testis to metagene 3, (Figure 2—figure supplement 3B). LgPCA can discern the differences in gene expression between certain samples (for example Brain and liver in PC2 of Figure 2). If it was intended to include the NMF analysis in this study, a point should be made as the rationale of which samples are to be clustered together and why, before the results may be compared to those from the lgPCA. Alternatively, the NMF analysis can be omitted since the lgPCA method is sufficient to accomplish the data analysis.

The analysis presented in Figure 1 compares the authors' data with that presented in the fetal datasets from the NIH Roadmap. The authors claim substantial up-regulation of sets of genes in their dataset but this is not quantified statistically (instead an arbitrary cut-off of >2-fold enrichment was employed). While there is relatively little data from which to perform a rigorous analysis it should be possible to use appropriate models (gaining power across genes such as GSEA) to find enriched categories in a more robust manner than presented here (i.e., this would allow appropriate correction for multiple testing).

---

## [Author Response]

Essential revisions:

1) There are some important issues raised by two of the reviewers with regard to the statistical analysis of the data. These concerns are provided in detail below and must be addressed in the revised manuscript.

Thank you. We have addressed these issues in turn in the relevant later section.

*2) The linkage of the transcriptional data to developmental disorders is not well validated. There is considerable mouse developmental and phenotype data available in databases that could be mined to enhance the value of your findings and help determine whether the genes identified are truly key regulators of tissue development.*

We agree that this is important. The signposting in the original submission was weak. We have addressed this by re-writing the main text to reference clearly [Supplementary-material SD1-data] (was F), which details 594 transcription factors at the extremes of principal components in the LgPCA. For each of these genes we mined the OMIM database for human developmental disorders (Column G of [Supplementary-material SD1-data]). To link to mouse developmental biology we mined the Mouse Genome Informatics (MGI) database for each of the 594 genes providing each MGI ID in Column I and consistent mouse phenotypic features (identifiable in 309 instances) in Column J. In addition, each gene was mined in Pubmed; where additional information from mouse models was particularly insightful over and above the MGI data PMIDs are given in Column K. These data reinforce the message that our identified transcription factors are indeed developmental regulators. In addition, it is relevant that the LgPCA led to consistent GO terms (e.g. Figure 2, ‘metanephros development’ for the embryonic kidney signature). These GO terms and the genes underlying them relate to regulatory roles discovered during mouse development.

*3) The claim that novel transcripts are likely lncRNAs was questioned and needs to be analyzed more carefully.*

We have debated this fast-moving area amongst ourselves throughout the course of the study. We considered it best to draw on up-to-date criteria and used the commentary by Mattick & Rinn in Nature Structural & Molecular Biology last year.

An important point is scale: we have discovered a huge amount of unannotated transcription and catalogued it as systematically as possible. This approach and publicising its existence is important because its scope (>6,000 transcripts) is beyond the capacity for functional investigation by any one lab. Thus, we view our job as to make the data open access for others to interrogate. This point was made by Mattick & Rinn: ‘Although these transcripts may be collectively referred to as long ‘noncoding’ RNAs (i.e. lncRNAs), this is a necessarily vague catch all term that is useful only as a moniker until they are better characterized’. We recognise, agree with and have adopted this initial starting point of using the ‘catch all term’, lncRNA. In addition, we have gone to great length to build on their suggested nomenclature (Figure 4 and [Supplementary-material SD1-data]) rather than pursue the easier option of simple ‘in-house’ numbering (prevalent in other manuscripts) which lacks reference to genomic structure and location and is less helpful to the community (‘Given the enormous numbers of long apparently noncoding transcripts, it is important to have a logical and flexible structure for cataloguing and parsing newly found transcripts, in a way that will foster rather than impede their functional and mechanistic exploration.’).

With this in mind: ‘The broad term lncRNA refers to a transcript >200 nt in length that does not appear to contain a protein-coding sequence.’ The 200bp threshold is regarded as a practical lower limit for standard RNA-seq thus excluding smaller RNAs, such as tRNAs, snRNAs and MIRs. We assessed coding potential using CPAT (Figure 4; Wang et al., NAR, cited in the manuscript), a method recommended by Mattick & Rinn. Our cut-off was <0.2, stricter than the 0.364 limit (determined by two-graph user operator characteristic) from the CPAT User Guide to increase our confidence that these transcripts are truly lncRNAs. Of note, over 90% of our novel transcripts had this very low coding potential.

We recognize that histone modifications can be useful for increasing the confidence with which transcriptional start sites of lncRNAs can be ascribed. We have started these studies (e.g. H3K4me3 ChIP-seq) for a number of organs as a part of a separate analysis of genome regulation (e.g. organ-specific enhancer usage during human embryogenesis). While too preliminary for inclusion here and well beyond the 2-month requested turnaround for this revision, it is reassuring that our early data on H3K4me3 marks do indeed correlate with the start of our lncRNAs.

Taken together, we feel that ‘lncRNA’ is the best descriptor for our novel transcripts filtered for length >200 bp and CPAT scores <0.2.

Statistical issues:

We note that the end-point of the Referees’ discussion on the NMF data is that we could simply remove it (which is a very helpful option—thank you!). We would like to reserve that opportunity; however, for now, we would like to try and preserve the NMF subject to the clarifications below for the following reasons. In previous advice we were encouraged to include an alternative approach that specifically targets organ-specific signatures. This seems reasonable. In addition, we feel that our manuscript will interest a broad audience not immediately familiar with the wide range of possible computational analyses. Therefore, it seems fair and reasonable to publicise the potential of NMF alongside our own approach of LgPCA. Clearly, if our clarifications and revisions are not convincing, the NMF analysis can be removed.

*There are concerns of the methodology that would require further clarification:*

*For the NMF, while the implementation of the methods is likely to be appropriate, there may be a potential flaw in the interpretation of what the analysis may achieved. Regarding "non-overlapping" metagenes, by default, genes can be represented in multiple metagenes, based on the Brunet algorithm in the NMF R package (also the default algorithm). The Methods state that this algorithm has been used, so following on from that, the assertion that "non-overlapping metagenes" could be extracted from the complete dataset and the "potential of coordinated deployment of overlapping genes" is ignored may be incorrect.*

Thank you for this point. Our original submission was not clear, some of the text needed moderation and we failed to reference [Supplementary-material SD1-data]. We applied the algorithm specifically to decipher tissue-defining gene-sets. Therefore, we used only those genes with ‘basis contribution’ >80% across metagenes meaning that each of the genes featured in only one metagene. This is depicted in Figure 1—figure supplement 5 and now referenced properly as [Supplementary-material SD1-data]. In not describing this adequately, we gave the false perception that this was the limit of the NMF approach when, as the reviewers correctly highlight, we could have reduced the threshold to allow overlapping sets of genes. We have amended to text to discuss this point and clarify our rationale.

*Furthermore, the critique that "NMF failed to discriminate transcriptional signatures for a number or organs or tissues, or discern the relationships between them… " may also apply to the outcome of lgPCA. Therefore, this is not a compelling argument for discarding the NMF.*

The original text was poorly chosen and we have tempered this section and the final paragraph. As a general aside, we feel that signatures were more comprehensively discerned in the LgPCA, in part by the opportunity to study profiles across PCs (e.g. adrenal gland across PC8 and PC9 as discussed in the text and as in [Supplementary-material SD1-data]). Moreover, we wanted to explore a methodology that permitted a lineage to be imparted upon the global dataset to model human embryogenesis. We have tried to get these points across in revised text.

*The analysis could have been taken further to examine the enrichment of all the metagenes, similarly to the one for liver (Figure 1—figure supplement 3) for all the specific tissue metagenes (“clear tissue-specific signals for thyroid, liver, RPE, brain, heart and adrenal gland”), and a similar approach could be taken for the downstream functional analysis of lgPCA.*

The inclusion of this figure panel and these data was in response to a helpful suggestion from a stem cell expert who commented on the importance of our data in allowing researchers to benchmark in vitro differentiated PSCs. In the example shown, we have taken the NMF metagenes (Figure 1—figure supplement 5) and tested for gene enrichment in data from Du et al. (2014, Cell Stem Cell) for human PSCs differentiated to hepatocytes with primary human adult hepatocytes (positive control) and human embryonic fibroblasts (negative control). The hepatocytes (both native and PSC-derived) from Du et al. showed appropriate signals in Metagene 2 (liver) but not the other metagenes. (It is interesting to note some signal for metagene 1 (thyroid – another anterior endoderm derivative) in the in vitro PSC differentiated hepatocyte-like cells.) This analysis required robust differentiation protocols and suitable RNA-seq data from differentiated PSCs; hence why we chose the exemplar of liver were data are available from Du and colleagues and hence why we feel it would be premature to extend the analyses to other NMF metagene or LgPCA organ signatures (lack of in vitro protocols/RNA-seq data).

For lgPCA, while the interpretation of the results would be correct, there is confusion in the nomenclature of loading and scores, which are fundamental concepts in PCA. PC scores refer to the eigenvalues (magnitude of variance) which determine the separation between the samples (as can be gleaned from Figure 2). PC loadings are eigenvectors and describe the contribution of the variables (in this case genes) that causes the separation of the samples. Therefore, PC scores should rightly be PCA loadings (main text, third and fourth paragraphs). This confusion should be attended to.

Thank you. We are very grateful for these important points. We have corrected this oversight.

Additionally, it would be helpful to mention at least once in the main text that when referring to "PC1 low" for example, this is meant to be low/negative PC1 scores.

We have added additional text.

*Both NMF and lgPCA are doing similar things, i.e. clustering samples based on gene expression to find which genes drive sample separation. The differences are that NMF can reveal the similarity between gene expression patterns for samples, which are clustered with the same metagenes, in this case kidney and testis to metagene 3, (Figure 2—figure supplement 3B). LgPCA can discern the differences in gene expression between certain samples (for example Brain and liver in PC2 of Figure 2). If it was intended to include the NMF analysis in this study, a point should be made as the rationale of which samples are to be clustered together and why, before the results may be compared to those from the lgPCA. Alternatively, the NMF analysis can be omitted since the lgPCA method is sufficient to accomplish the data analysis.*

We have adapted the text. NMF can indeed generate metagenes across tissues (e.g. kidney and testis as pointed out) as can LgPCA (e.g. limbs and palate in PC3). As commented on, LgPCA can also discern differences in gene co-expression between organs (e.g. tongue and heart in PC15). To note, NMF does not start from a position of pre-determined sample clustering. Hoping to resolve this issue and recognising that the original text was unhelpful on NMF versus LgPCA, we have made revisions. As above, if these measures are inadequate, we can always remove the NMF.

*The analysis presented in Figure 1 compares the authors' data with that presented in the fetal datasets from the NIH Roadmap. The authors claim substantial up-regulation of sets of genes in their dataset but this is not quantified statistically (instead an arbitrary cut-off of >2-fold enrichment was employed). While there is relatively little data from which to perform a rigorous analysis it should be possible to use appropriate models (gaining power across genes such as GSEA) to find enriched categories in a more robust manner than presented here (i.e., this would allow appropriate correction for multiple testing).*

Thank you for this valid point. We are grateful for the Referees noting the relative paucity of data for this analysis. Nevertheless, we feel that it is a useful early component of the paper to show that there are indeed differences between human embryonic datasets and those from later development from NIH Roadmap. We have undertaken alternative analyses based on statistical cut-offs and with methodology that facilitates multiple testing. These data are now included in the manuscript in place of the previous Figure 1.